# The Impact of Beeswax and Glycerol Monolaurate on Camellia Oil Oleogel’s Formulation and Application in Food Products

**DOI:** 10.3390/molecules29133192

**Published:** 2024-07-04

**Authors:** Xingchen Wei, Ronghui Xia, Chenxi Wei, Longchen Shang, Jianhui An, Lingli Deng

**Affiliations:** 1College of Biological and Food Engineering, Hubei Minzu University, Enshi 445000, China; 202330385@hbmzu.edu.cn (X.W.); 072240148@hbmzu.edu.cn (R.X.); 072240160@hbmzu.edu.cn (C.W.); 2Hubei Key Laboratory of Selenium Resource Research and Biological Application, Hubei Minzu University, Enshi 445000, China; 2021021@hbmzu.edu.cn; 3Hubei Key Laboratory of Biological Resources Protection and Utilization, Hubei Minzu University, Enshi 445000, China

**Keywords:** camellia oil, fatty acid, oleogel, beeswax, glyceryl monolaurate, biscuit, sausage

## Abstract

This study assessed the nutritional profile of camellia oil through its fatty acid composition, highlighting its high oleic acid content (81.4%), followed by linoleic (7.99%) and palmitic acids (7.74%), demonstrating its excellence as an edible oil source. The impact of beeswax (BW) and glycerol monolaurate (GML) on camellia oil oleogels was investigated, revealing that increasing BW or GML concentrations enhanced hardness and springiness, with 10% BW oleogel exhibiting the highest hardness and springiness. FTIR results suggested that the structure of the oleogels was formed by interactions between molecules without altering the chemical composition. In biscuits, 10% BW oleogel provided superior crispness, expansion ratio, texture, and taste, whereas GML imparted a distinct odor. In sausages, no significant differences were observed in color, water retention, and pH between the control and replacement groups; however, the BW group scored higher than the GML group in the sensory evaluation. The findings suggest that the BW oleogel is an effective fat substitute in biscuits and sausages, promoting the application of camellia oil in food products.

## 1. Introduction

Solid fats are extensively used in the food industry for desirable attributes, such as pleasing taste, superior texture, stability, and pliability. However, they contain trans- and saturated fatty acids, which are linked to diabetes, obesity, cancer, and cardiovascular diseases [1]. Numerous fat substitutes, including the ones based on carbohydrates and proteins, have been developed [2,3], which can emulate the texture of fats to some degree, but often fail to replicate the taste and flavor of conventional foods. Oleogelation is an innovative method that structures oils by integrating edible oleogel molecules into liquid oil without changing its original chemical composition. Oleogels can mimic the characteristics of solid fats, such as rheological properties, viscoelasticity, spreadability, and hardness, and typically have a lower content of saturated fatty acids [4]. By adding a gelling agent to liquid oil, oleogels obtained a three-dimensional network through hydrogen bonds and van der Waals forces during the cooling of the gelling agent, which encapsulates the vegetable oils, reducing their fluidity and creating a viscoelastic solid fat compound [5]. Despite the loss of fluidity, oleogels maintain the original properties of liquid oils and exhibit solid fat-like characteristics, with a substantial amount of liquid oil remaining [6]. Consequently, oleogels serve as suitable replacements for traditional fats, providing a low-saturated, zero-trans-fat alternative for producing baked foods [7], cheese [8], chocolate [9], and ice cream [10].

Camellia oil, along with olive oil, coconut oil, and palm oil, is recognized as one of the four principal woody plant oils globally [11]. As a traditional vegetable oil, camellia oil has a history of consumption over a thousand years in China [12]. The profile of camellia oil is close to olive oil, particularly in terms of its fatty acid composition. This similarity has earned camellia oil the name “Eastern Olive Oil” [13], confirming it as a nutritious and healthful edible oil. Camellia oil contains a high content of unsaturated fatty acids (oleic acids, linoleic acids, linolenic acid, etc.) [14] and natural antioxidants (squalene, phytosterols, polyphenols, lipid-soluble vitamins, etc.) [15], conferring strong antioxidant properties. They prevent hyperglycemia, hypertension, arteriosclerosis, and skin aging [16]. Dispersing various gelling agents to camellia oil for oleogel formation involves no chemical reactions, preventing the formation of trans fats. The resulting oleogels are rich in unsaturated fatty acids, offering a healthy fat option [17]. Here, beeswax (BW) and glycerol monolaurate (GML) were selected as gelling agents. BW was used owing to its gel-forming properties at low concentrations; wax esters and long-chain fatty alcohols crystallize, creating a network that retains liquid oil [18]. GML is widely used in the food industry because it acts as an emulsifier and has preservative qualities, and is cost-effective and safe [19]. GML crystallizes to form a tight three-dimensional network in oleogels, and its addition to oleogels strengthens gel structure, thermal stability, and ability to resist deformation [20].

This investigation examined the fatty acid profile of camellia oil and utilized it as a foundation for the development of oleogel systems. Utilizing BW and GML as gelling agents, a series of oleogel formulations with varying concentrations were constructed. To contrast the difference in gelation characteristics between them, color, texture, and structural analyses of camellia oil oleogels were conducted to develop a superior olegel system, which would provide insights into the formulation of oleogels in general. Then camellia oil oleogels were integrated as fat substitutes in the preparation of biscuits and sausages, which aimed to assess the viability of camellia oleogels as an alternative to conventional fats in food products, potentially enhancing the nutritional profile and texture of these items. By evaluating the adaptability of camellia oleogels in diverse foods, this work contributes to the advancement of healthier and innovative food product development.

## 2. Results and Discussion

### 2.1. Fatty Acid Composition of Camellia Oil

The fatty acid composition is a crucial parameter for evaluating the quality of edible oils. As shown in Table 1, camellia oil contains tridecanoic acid (C13:0), palmitic acid (C16:0), palmitoleic acid (C16:1), stearic acid (C18:0), oleic acid (C18:1), linoleic acid (C18:2), linolenic acid (C18:3), arachidic acid (C20:0), and eicosenic cis acid (C20:1). Among these, oleic acid has the highest content at 81.37%, followed by linoleic acid at 7.99% and palmitic acid at 7.74%. The contents of other fatty acids are between 0–2%. Oleic acid and linoleic acid make up the largest proportion of fatty acids in camellia oil, consistent with previous studies [21]. Camellia oil shares a comparable fatty acid profile with olive oil characterized by a high content of oleic acid (ranging from 59.78% to 79.07%), along with significant proportions of linoleic acid (3.69% to 18.73%) and palmitic acid (9.25% to 16.47%) [22]. Thus, camellia oil can be considered a highly nutritious oil.

### 2.2. Effect of Gelling Agents on Camellia Oil Olegels

#### 2.2.1. Formation and Color of Camellia Oleogels with Different Concentrations of Gels

The results indicate that minimums of 3% BW and 2% GML are required to successfully form an oleogel. At concentrations exceeding 3% BW and 2% GML, all samples exhibited a semisolid structure and did not flow upon the inversion of test tubes.

The color of the oleogel influences its suitability for use in the food industry. Table 2 and Table 3 demonstrate that adding BW and GML significantly alters the color of camellia oleogels (*p* < 0.05). The L* value, indicating brightness, increased markedly with the addition of BW and GML, peaking at 10% in both cases. It seems that the less refraction of light through more compact structures, which are created as a result of a higher concentration of gelator, produces a lighter color [23]. In BW-formed oleogels, negative a* and b* values indicated a green–yellow hue, because of the inherent color of BW. This hue may be influenced by the presence of wax esters in BW, with amounts of plant pollens containing yellow-colored wax-soluble compounds [24]. In GML-formed oleogels, a negative a* and a positive b* value indicate a slight green–blue hue. These findings align with the visual color changes depicted in Figure 1 for the camellia oleogel.

#### 2.2.2. Texture Determination of Camellia Oleogels

The hardness and springiness of oleogels are important indicators of their potential application in food [25]. These properties are influenced by the type and amount of gel additives used. The hardness of an oleogel refers to its resistance to deformation under an applied force. Springiness describes the ability of an oleogel to return to its original shape post-deformation [26]. A balance between hardness and springiness is often sought to create oleogels that are strong and flexible.

The data presented in Table 4 and Table 5 reveal distinct textural attributes between oleogels formulated with BW and GML. A significant variation in hardness was observed among the BW oleogels, with values spanning from 0.4573 N to 10.5710 N, while the hardness of GML oleogels ranged from 0.0703 N to 0.4521 N. This range indicated the influence of types and concentrations of gels on the mechanical properties of the oleogels. As the gel content increased, the hardness and springiness of both oleogels increased significantly, suggesting that higher concentrations of gelling agents making the gels more rigid [27]. The hardness of BW oleogels was significantly higher than GML oleogels at the same dosage, primarily attributed to the distinct gelation mechanisms of the two types of gels [28]. Hwang’s findings [29] suggested that a denser crystal network correlated with the increase in oleogel hardness, thus it was plausible that BW formed a more robust crystalline microstructure than GML.

#### 2.2.3. Infrared Spectroscopy of Camellia Oleogels

To investigate the molecular interactions within the oleogel system, the absorption peaks of its molecular characteristics were analyzed using infrared spectroscopy. The FTIR spectra of camellia oleogels are depicted in Figure 2. The spectra shows specific peaks at approximately 2921, 2850, 1743, 1460, 1160, and 720 cm^−1^. The strong peak at 1743 cm^−1^ indicates the carbonyl (C=O) stretching vibration of triglycerides, signifying the presence of glycerol and fatty acids linked by ester bonds [30]. The peaks at 2921 cm^−1^ and 2852 cm^−1^ correspond to the stretching vibrations of CH2 groups in the aliphatic chains. In oleogels prepared with BW, with the increase in BW concentrations, the peaks at 2921 cm^−1^ and 2852 cm^−1^ shifted slightly to lower wavenumbers, because van der Waals interactions were involved in the construction of oleogels [31]. The peak at 1160 cm^−1^ was mainly due to the stretching vibrations of the C-O bond, associated with ether or ester bonds, revealing details of intermolecular interactions [32]. The addition of BW and GML did not significantly change the peak positions, indicating the chemical structure of the oil molecules remained unchanged, and no new chemical groups formed. These findings suggest that the oleogel structure is formed by non-covalent interactions, such as hydrogen bonds and van der Waals forces [33]. The results imply that the formation of the oleogel does not rely on the formation of chemical bonds, but rather on enhancing the physical interactions between molecules to stabilize its structure.

### 2.3. Properties of Biscuits Prepared Using Various Fat Types

#### 2.3.1. Texture and Color of Biscuit Dough

Biscuit dough made with different types and amounts of fats exhibited significant differences in texture (Table 6). The texture experiment examined differences in four aspects: hardness, springiness, resilience, and stickiness. The dough with 6.7% BW had the highest hardness (25.24 ± 0.38 N), potentially due to the crystallization pattern of BW. This was followed by the butter control group (21.99 ± 1.05 N). The GML groups depicted lower hardness, consistent with the hardness of GML oleogels. Among the five experimental groups, there was no significant difference in springiness, with all values falling within the same range. For resilience, the 6.7% GML groups was closest to the control group. Concerning stickiness, the butter control and 10% BW groups had no significant difference. Thus, the type and amount of fat used in biscuit dough significantly affected its texture characteristics. BW, especially at 6.7%, increased the hardness and stickiness, whereas 10% GML increased the resilience. Using BW-based oleogels, especially at 6.7%, resulted in dough with rheological properties similar to butter-based dough. In terms of color, only the L* value showed significant differences among the five dough groups, indicating that the experimental groups partially affected the dough’s color. Therefore, oleogels can be used as an alternative to solid fat in cookies [34].

#### 2.3.2. Physical Properties of Biscuits

The texture characteristics, expansion ratio, and color parameters of biscuits formulated with different fats were analyzed (Table 7). The fats tested included butter, two concentrations of GML (6.7% and 10%), and two concentrations of BW (6.7% and 10%).

The texture of biscuits was determined using the three-point bending method, where a distinct peak in the curve during the first compression indicated brittleness, and the second peak represented hardness. A single peak indicated that the hardness and brittleness of the biscuit were the same [35]. The hardness of the biscuits was significantly influenced by the type and concentration of the fat used. Biscuits made with 10% BW exhibited the highest hardness (30.58 ± 3.17 N), indicating a significantly firmer texture than other formulations [36] and may potentially be related to the microstructure of the oleogel and intermolecular forces [37]. It was consistent with the hardness change in the BW oleogel mentioned above. Conversely, biscuits made with 6.7% GML demonstrated the lowest hardness (13.87 ± 0.38 N), suggesting a soft texture. Resilience, which measures the ability of the biscuit to return to its original shape after deformation [38], was highest in the 6.7% GML formulation (0.53 ± 0.16). By contrast, biscuits with 10% BW had the lowest resilience (0.25 ± 0.068).

The expansion ratio, indicating an increase in biscuit volume during baking, peaked at 10% GML (338.37 ± 16.69) and was lowest at 10% BW (192.8 ± 3.76). An excessively high expansion ratio resulted in crust shrinkage, diminishing crispness [39]. The results reveal that the two groups of biscuits made with BW oleogels have expansion ratios similar to those made with butter, suggesting that BW oleogel is better suited for fat replacement in biscuits.

Color parameters L*, a*, and b* describe the lightness and color tone of the biscuits. Lightness (L*) was the highest for biscuits made with 6.7% GML (76.42 ± 0.22), indicating a lighter product. Biscuits made with 6.7% BW were the darkest (L* = 60.76 ± 1.178). The a* value—which indicates redness—was highest for biscuits made with 6.7% BW (14.09 ± 1.60), and lowest for biscuits made with 6.7% GML (0.78 ± 0.12). The b* value, representing yellowness, was highest for biscuits made with 10% BW (35.65 ± 1.55) and lowest for biscuits made with butter (25.57 ± 1.14).

Fat type and concentration significantly affected biscuit texture, expansion, and color. GML at a 10% concentration provided a good balance of softness, high expansion ratio, and desirable color, while BW—particularly at 10%—resulted in a harder, crunchier texture with significant color changes.

#### 2.3.3. Sensory Evaluation of Biscuits

As shown in Figure 3, the appearance scores for the BW, 6.7% GML and butter control groups are all close to 7 points. The 10% GML group scored the lowest and exhibited slight oil seepage. For color, the 10% BW and 10% GML groups scored 6.375 points, whereas the 6.7% GML group scored the lowest (5.375 points). In texture, the 10% GML group scored the lowest at 5 points, whereas the other four groups had similar points. For taste, the two BW groups achieved higher scores at approximately 6.75, whereas the 10% GML group had the lowest score at 4. For saltiness, the 6.7% BW group scored the highest with 7.125 points, and the 10% GML group scored the lowest with 5 points. Overall, the 10% BW group had the highest score at 6.875, while the 10% GML group had the lowest score at 4.75 because of a strong odor that affected the sensory experience.

Among the five groups, the 10% BW group was the most favored, followed by the 6.7% BW. The results indicate that beeswax is a more suitable substitute for fat in the form of oleogels in biscuits.

### 2.4. Application of the Oleogels When Preparing Sausages

#### 2.4.1. Influence of Different Fats on the Texture of Sausages

The control group exhibited greater hardness, chewiness, and resilience than the butter and oleogel substitution groups (Table 8). The hardness of the samples varied significantly across different fat replacements. The order of hardness was control (11.92 ± 0.20 N) > butter (11.25 ± 0.13 N) > BW (10.23 ± 0.24 N) > GML (6.70 ± 0.31 N). The chewiness of the samples followed a similar trend. There were no significant differences in the resilience, springiness, and cohesiveness between the control and substitution groups. The results demonstrate that the type of fat replacement significantly affects the textural properties of sausage. Butter and BW showed similar textural properties to the control but had slightly lower values. However, GML resulted in significantly lower hardness, chewiness, resilience, and cohesiveness, indicating a markedly different texture. This was related to the crystalline form of the gelling agent [20]. GML forms a cyclic crystalline network through weak hydrogen bonds, which prevents the flow of liquid oil and forms a relatively weak oleogel [40]. Consequently, sausages prepared with GML as a fat substitute exhibited the lowest textural property values among all samples. By contrast, BW changed n-alkanes or wax esters into microplates, forming a crystalline particle spatial network with good gelling properties [41]. BW can effectively replace fat in sausages, resulting in textural properties closer to those of the control group. Overall, BW is a promising alternative to lard for maintaining desirable textural properties in sausage, whereas GML may require further optimization for similar applications.

#### 2.4.2. Water Holding Capacity of Sausages

Water holding capacity is a crucial indicator of meat product quality, measuring the ability of protein gels to bind water [42]. Figure 4 shows no significant difference in the water holding capacity between the control and gel substitute groups. Although the BW group had a lower capacity, it was not significantly different from the control group.

#### 2.4.3. Color and pH of Sausages

During the preparation of the experimental samples, no coloring agents were added to avoid any unrelated effects. Table 9 shows no significant differences in brightness, yellowness, and redness between the control and substitute groups, indicating that the substitutes did not affect the inherent brightness of the sausage. The BW group had a relatively high yellowness value (15.10 ± 2.61), possibly owing to the natural color of BW. There were no significant differences in pH between the control and substitute groups, suggesting that the gel substitutes had a minimal effect on acidity.

#### 2.4.4. Sensory Assessment of Sausages

The sensory evaluation of sausage samples, with varying fat supplements, is presented in Figure 5. Overall, the butter-treated group most closely resembles the control group, with a score of approximately 89.4, whereas the control group scored 92.2. For appearance, the GML group received the lowest score because of the observed oil and water seepage. For color, no significant differences were found across the four groups; this was consistent with the data in Table 9. Concerning flavor, taste, and organizational status, the BW group scored higher than the GML group because the GML group had a looser structure and an undesirable taste. Overall, the BW oleogel was a suitable fat substitute in sausage.

## 3. Materials and Methods

### 3.1. Materials

Camellia oil, GML (food grade), and BW (food grade) were supplied by Zhejiang Shanshen Camellia Oil Development Co., Ltd. (Quzhou, China) and Hangzhou Kangyuan Food Technology Co., Ltd. (Hangzhou, China). Fatty acid standard samples, n-hexane, and carbinol were of chromatographic grade, purchased from Sigma-Aldrich (Shanghai, China). Potassium hydroxide (KOH) and sodium bisulfate (NaHSO_4_) were of analytical grade, purchased from Sinopharm Chemical Reagent Co., Ltd. (Shanghai, China).

### 3.2. Analysis of Fatty Acid in Camellia Oil

A total of 0.06 g of camellia oil was placed into an esterification vial for methylation using the potassium hydroxide–methanol method, as described in [43]. To the vial, 5 mL of n-hexane was added, and the mixture was shaken for 5 min for dissolution. Subsequently, 0.2 mL of a 2 mol/L KOH methanolic solution was introduced and the vial was shaken for an additional 5 min to facilitate the reaction. After the reaction, the upper layer of the solution was carefully transferred into a clean bottle for next step. To this, 1 g of sodium bisulfate (NaHSO_4_) was added, and the mixture was shaken for 1 min to neutralize the KOH, resulting in the precipitation of the salt. The supernatant was then collected and used for *GC* analysis by using a gas chromatograph (Agilent 7890B, Santa Clara, CA, USA). The GC conditions were as follows [44]: automatic sampler; FID detector; Agilent DB-23 column (30 m × 0.25 mm × 0.2 μm); the carrier gas was helium with a flow of 1 mL/min; the injection volume was 1 μL; the split ratio was 1:50; heating program: 100 °C (initial temperature, 3 min retention), ramped to 230 °C (10 °C/min retention for 8 min), with a total run time of 30 min; injection port temperature: 250 °C; and detection temperature: 250 °C. The fatty acid composition was determined by comparing the retention time with various fatty acid standard samples [45]. The relative content of each fatty acid was calculated using the peak area normalization method.

### 3.3. Camellia Oil Oleogels’ Preparation

The preparation of the oleogels was based on the method of Qu, K [26], with slight modifications. In total, 15 g of camellia oil was placed in glass bottles and mixed with incremental percentages of BW: 1% (0.15 g), 2% (0.3 g), 3% (0.45 g), 4% (0.6 g), 5% (0.75 g), 6% (0.9 g), 7% (1.05 g), 8% (1.2 g), 9% (1.35 g), and 10% (1.5 g), labeled B1–B10, respectively. Similarly, 15 g of camellia oil was placed in glass bottles and mixed with incremental percentages of GML: 1% (0.15 g), 2% (0.3 g), 3% (0.45 g), 4% (0.6 g), 5% (0.75 g), 6% (0.9 g), 7% (1.05 g), 8% (1.2 g), 9% (1.35 g), and 10% (1.5 g), labeled G1–G10, respectively. To prepare the oleogels, the mixed camellia oil sample was heated at 90 °C for 120 min using a thermostatic magnetic stirrer at a rate of 500 rpm/min (S10-3, Shanghai Sile Instruments Co., Ltd., Shanghai, China) until fully dissolved. Then, each hot mixed solution was transferred to a 4 °C refrigerator (BCD-601WDPR, Haier Electric Appliance, Qingdao, China) for 24 h. After that, each sample was stored in a biochemical incubator (SPX-150BSH-II, Shanghai CIMO Medical Instrument Co., Ltd., Shanghai, China) at 25 °C for further research.

#### 3.3.1. Colorimetric Analysis of Camellia Oil Oleogels

The prepared camellia oil oleogels were subjected to a controlled heating process in a water bath (C-HH-6, Beijing Ruicheng Yongchuang Technology Co., Ltd., Beijing, China), targeting a temperature of 90 °C to facilitate complete dissolution. When the oleogels were fully liquefied, a volume of 3 mL was transferred into individual wells of a 24-well plate. Subsequently, the plate was allowed to equilibrate at room temperature to encourage the gel to re-solidify, a process that typically spans approximately 2 h. Then the re-solidified samples were assessed using Color Spectrum (CS-820N, Hangzhou CHNS pec Technology Co., Ltd., Hangzhou, China), with each sample group measured in triplicate. Color variations in camellia oil oleogels with different gelling agent concentrations were evaluated using L*, a*, and b*, indicating brightness, red–green contrast, and yellow–blue contrast [46].

#### 3.3.2. Texture Analysis of Camellia Oleogels

The hardness and springiness of the oleogels were measured using Texture Analyzer (Shanghai Baosheng Industrial Development Co., Ltd., Shanghai, China) at 25 °C. Prepared camellia oleogel samples were positioned under the instrument probe, ensuring flat contact with the testing surface of the probe. The parameters were as follows: pre-test speed was 2.0 mm/s, test speed was 1.0 mm/s, and post-test speed was 5.0 mm/s. A cylindrical TA/0.5 gel probe was employed, with a 15 mm compression depth and 5 gf trigger force [47].

#### 3.3.3. Infrared Spectroscopy Detection of Camellia Oleogels

Infrared spectroscopy (FTIR-ATR, Thermo, Waltham, MA, USA) was used to analyze the molecular structure of camellia oleogels with different amounts of gelling agents. The infrared spectrum was acquired through scanning from 400 to 4000 cm^−1^ at a resolution of 4 cm^−1^ and a temperature of 20 °C [31].

### 3.4. Application of Camellia Oleogel to Replace Biscuit Fat

#### 3.4.1. Preparation of Biscuits

The biscuits comprised 150 g of wheat flour, 30 g of fat, 1 g of baking soda, 2 g of salt, 3 g of yeast powder, and 60 mL of milk [48]. In total, 150 g of wheat flour was mixed with 1 g of baking soda and 2 g of salt, and 60 mL of milk was gradually stirred in. Subsequently, 30 g of fat was added to the mixture. The dough underwent a kneading process for a duration of 20 min, followed by fermentation at 25 °C in a fermentation cabinet (DHTHM-16-0-P-SD, Doaho Test Co., Ltd., Shanghai, China) for 30 min. After fermentation, the dough was rolled out to thickness of 1 cm, and shaped using a mold. The oven was preheated to 170 °C. A baking tray was lined with absorbent paper. The dough shapes were spaced evenly on the tray to ensure even baking and prevent sticking; it was baked for 20 min. The baked biscuits were cooled, packed, and labeled. Drawing from the conclusions of our preceding experiments, it was observed that the oleogels’ formulation containing 10% (1.5 g) of the gel agent exhibited significant differences in properties. So, the 10% concentration was selected for further investigation. Furthermore, to explore the effect of oleogels formed by lower gel agent concentrations on food products, a formulation containing 6.7% (1.0 g) of the gel agent was also chosen for study. Consequently, the fats used included butter, and camellia oleogels at 6.7% and 10% BW, as well as 6.7% and 10% GML, with all other conditions held constant.

#### 3.4.2. Texture Measurement of Dough

After fermentation, 150 g of dough was determined for texture measurement using a 6 mm diameter cylindrical probe for puncture tests. The parameters were as follows: pre-test speed was 3 mm/s, test speed was 1 mm/s, and post-test speed was 1 mm/s, with a 20 mm displacement and 5 gf trigger force. Each sample was tested thrice.

#### 3.4.3. Texture Measurement of Biscuits

The cooled biscuit samples were stored in hermetically sealed bags for evaluation at room temperature for 0–3 days. A three-point bending fracture probe was employed for comprehensive biscuit texture analysis, using a pre-test speed of 3 mm/s, a post-test speed of 3 mm/s, and a test speed of 1 mm/s. Each biscuit sample was tested thrice.

#### 3.4.4. Measurement of Biscuit Expansion Ratio

The baking expansion ratio of the biscuits was characterized by the change in biscuit thickness before and after baking. The biscuit expansion ratio was expressed as X = h_2_/h_1_ × 100%, where X is the expansion ratio in percent, h_1_ is the thickness of unbaked dough in millimeters, and h_2_ is the thickness of baked biscuit in millimeters [35].

#### 3.4.5. Color Measurement of Dough and Biscuits

The color characteristics of the dough and biscuits were assessed using a Color Spectrum (CS-820N, Hangzhou CHNS pec Technology Co., Ltd., Hangzhou, China). Prior to the measurement, the device was calibrated, and the prepared dough samples were shaped into uniform sheets using a mold, with dimensions of 5 cm in length, 4 cm in width, and 1 cm in thickness. Similarly, the biscuits, once cooled to room temperature, were also measured for their color attributes. Each sample was analyzed in triplicate to ensure the reliability of the results. The color measurements were conducted using the L*, a*, and b* color systems: the L* value, indicative of lightness, scales from 0 (black) to 100 (white); the a* value, which represents the red–green axis, varies from negative values (green) to positive values (red); and the b* value, representing the yellow–blue axis, ranges from negative (blue) to positive (yellow). These parameters were determined based on three independent analyses, as referenced in the literature [49].

#### 3.4.6. Sensory Evaluation of Biscuits

Biscuits made with various fats were assessed for sensory quality, each labeled with a unique random number. A nine-point scoring hedonic scale system was applied, where one indicated maximum dislike, five indicated neither like nor dislike, and nine indicated maximum appreciation [50]. Samples were rated based on appearance, color, texture, taste and salinity, and the final score was determined by averaging these ratings.

### 3.5. Application of Camellia Oleogel to Replace Sausages Fat

#### 3.5.1. Preparation of Sausages

The sausages contained 88 g of pork tenderloin, 16 g of corn starch, 4 g of salt, 2 g of white sugar, 0.4 g of monosodium glutamate, 40 g of ice water, and 0.36 g of pepper. The control group contained 36 g of lard. The treatment groups contained 18 g of lard + 18 g of butter; 18 g of lard + 18 g of 10% BW camellia oleogel; and 18 g of lard + 18 g of 10% GML camellia oleogel.

The sausages were prepared by trimming visible fat and connective tissue from fresh the pork tenderloin and cutting it into small strips. The pork strips were mixed with salt, monosodium glutamate, and 1/3 of the ice water, and minced in a meat grinder for 20 s. Next, corn starch, white sugar, and 1/3 of the ice water were minced with the mix for 15 s. Finally, fat, oleogel, spices, and the remaining ice water were stirred in until the mixture was smooth, uniform, and glossy. The mold was filled with the prepared mixture, placed in a steamer at 100 °C for 20 min, and cooled.

#### 3.5.2. Texture Analysis of Sausages

Sausage samples were prepared, measuring 1 mm in height and 25 mm in diameter. A Texture Analyzer with a P50 probe was used for full texture analysis. The parameters were as follows: pre-test speed of 2.0 mm/s, test speed of 0.8 mm/s, and post-test speed of 0.8 mm/s, with a determination time interval of 5 s and a trigger force of 5 g, and a compression ratio of 25%.

#### 3.5.3. Water Holding Capacity Determination

Sausage samples were cut into approximately 1 cm^2^ pieces, weighed, placed in centrifuge tubes, centrifuged at 6000 r/min for 15 min at 25 °C, dried with filter paper, and re-weighed [51].
Water-Holding Capacity (%) = M2/M1 × 100
where M1 and M2 are the masses of sausage samples before and after water absorption, respectively.

#### 3.5.4. Color Measurement

A colorimeter was used to measure the cross-section of the sausage, with D65 as the light source and a standard white plate for calibration. The results are expressed concerning brightness (L*), redness (a*), and yellowness (b*).

#### 3.5.5. pH Determination

During sausage preparation, pH was determined with a pH meter on homogenates of minced meat in water in 1:10 (*w*/*v*) ratio.

#### 3.5.6. Sensory Evaluation

A panel of 15 students majoring in food science was recruited for sensory analysis. The sausages were prepared by steaming at 100 °C for 20 min. Post-cooking, the samples were presented to the participants immediately, each labeled with a unique random number. Prior to the sensory analysis, the objectives and method of the evaluation were introduced, including the use of standardized scales. The sensory attributes assessed included the appearance, color, flavor, taste, and organizational status of the sausage samples. Each sensory attribute was evaluated using a 20-point scale, contributing to a cumulative total of 100 points for the overall sensory evaluation.

### 3.6. Statistical Analysis

All tests were measured at least in triplicate. Data are presented as mean ± standard deviation, and Origin 8.0 was used for analysis. Means were analyzed using the analysis of variance (ANOVA). Significance was defined as *p* < 0.05.

## 4. Conclusions

Camellia oil, abundant in oleic and linoleic acids, formed oleogels with BW or GML that significantly influence the system’s physicochemical properties through enhancing molecular interactions. The oleogel with 10% GML exhibited the highest hardness and springiness, suggesting its potential as a solid fat substitute. During biscuit production, the GML oleogel was superior for crispness and texture, but scored lower in the sensory analysis because of odor issues. By contrast, the BW oleogel was favored for its sensory attributes, indicating its suitability as a fat substitute in biscuits. In sausages, although hardness varied in samples, other properties, such as resilience, springiness, cohesiveness, water holding capacity, color, and pH, were consistent. The sensory evaluation favored the BW oleogel over the GML oleogel. Overall, the BW oleogel serves as a suitable fat substitute for preparing biscuits and sausages, offering health and nutritional benefits. These findings are significant for advancing oleogel innovation in the food industry and promoting camellia oil utilization.

## Figures and Tables

**Figure 1 molecules-29-03192-f001:**
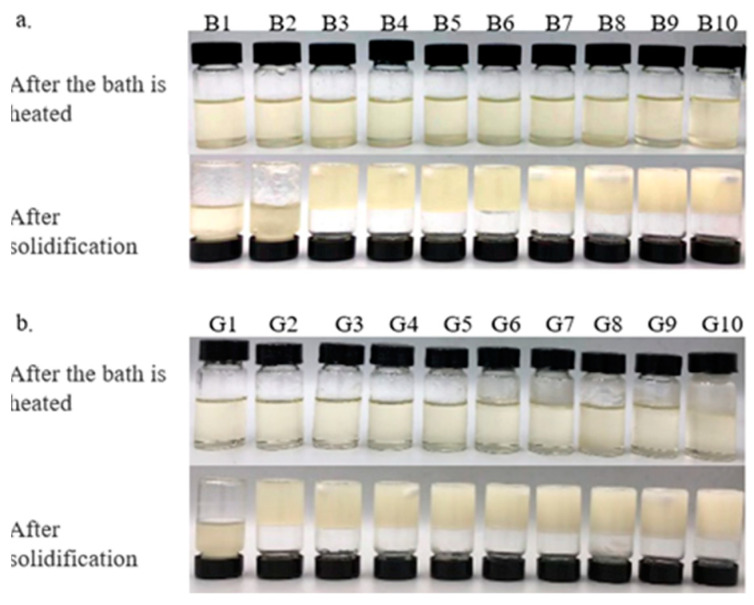
Camellia oil oleogels with different concentrations of beeswax (BW) and glycerol monolaurate (GML): (**a**)-oleogels were formed by BW, (**b**)-oleogels were formed by GML.

**Figure 2 molecules-29-03192-f002:**
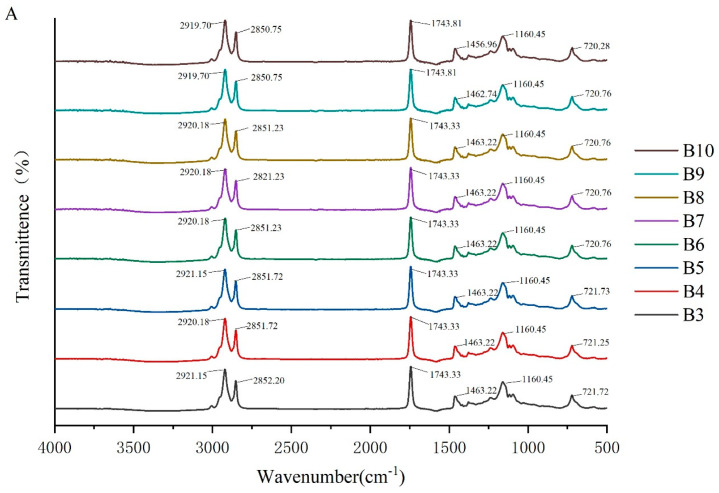
FTIR spectra of camellia oleogel prepared with different concentrations of BW (**A**) and GML (**B**).

**Figure 3 molecules-29-03192-f003:**
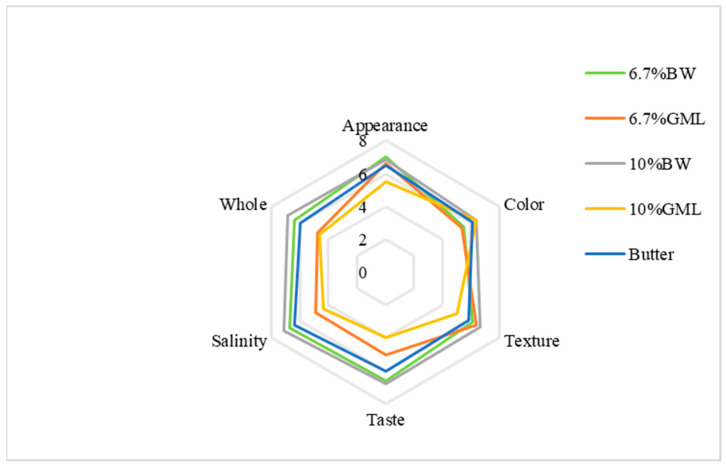
Sensory evaluation analysis of biscuits prepared with different fats.

**Figure 4 molecules-29-03192-f004:**
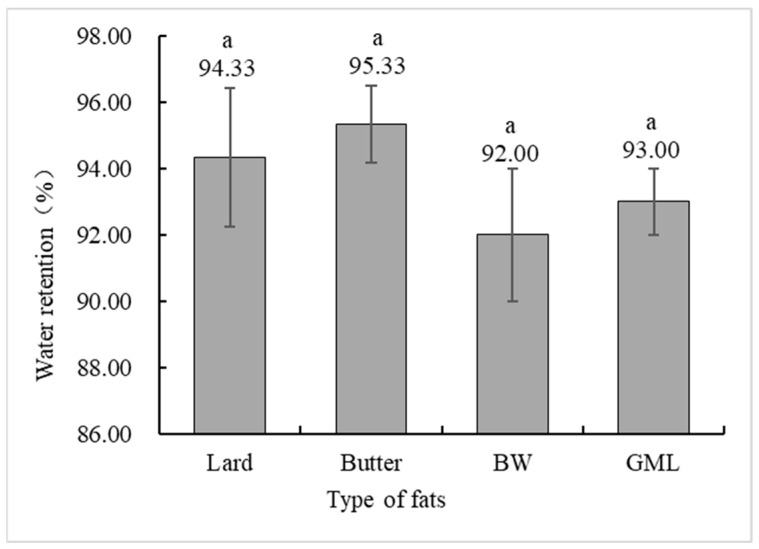
Water holding capacity analysis of sausages made with different types of fats. Note: Lard: each formulation contains 36 g of lard. Butter: each formulation contains 18 g lard + 18 g butter. BW: each formulation contains 18 g lard + 18 g 10% BW camellia oleogel. GML: each formulation contains 18 g lard + 18 g 10% GML camellia oleogel. Letters in the same row indicate significant differences (*p* < 0.05).

**Figure 5 molecules-29-03192-f005:**
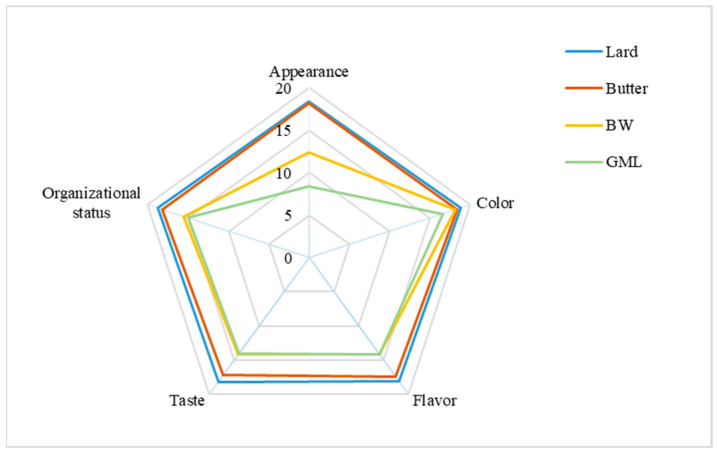
Sensory evaluation analysis of sausages prepared with different fats.

**Table 1 molecules-29-03192-t001:** Fatty acid composition of camellia oil.

Fatty Acid		Retention Time	Peak Area	%wt/ Fatty Acid Content
tridecanoic acid	C13:0	13.7335	7.15	0.1395
palmitic acid	C16:0	16.7260	398.55	7.7405
palmitoleic acid	C16:1	17.0710	4.40	0.0855
stearic acid	C18:0	19.7475	87.95	1.7090
oleic acid	C18:1n9c	20.3250	4189.85	81.3710
linolelaidic acid	C18:2n6t	20.9235	7.30	0.1420
linoleic acid	C18:2n6c	21.1115	403.90	7.8435
linolenic acid	C18:3n3	22.2030	11.65	0.2265
arachidic acid	C20:0	23.3990	18.20	0.3540
eicosenic cis acid	C20:1	23.7840	20.00	0.3885

Note: %wt represents weight content. The components with the C number before the colon represent total number of carbons, while the numbers after the colon represent the total number of double bonds; n-represents the position of the first double bond counting from the methyl end or omega end.

**Table 2 molecules-29-03192-t002:** Color of camellia oleogels prepared with different concentrations of BW.

Sample	L*	a*	b*
3% (B3)	30.00 ± 0.91 ^e^	−0.49 ± 0.05 ^a^	−0.82 ± 0.20 ^a^
4% (B4)	31.60 ± 0.75 ^e^	−0.72 ± 0.21 ^a^	−2.00 ± 0.39 ^b^
5% (B5)	34.05 ± 1.14 ^d^	−1.41 ± 0.35 ^b^	−3.97 ± 0.56 ^bc^
6% (B6)	36.97 ± 0.80 ^c^	−1.71 ± 0.08 ^b^	−4.24 ± 0.37 ^cd^
7% (B7)	39.02 ± 0.31 ^c^	−1.60 ± 0.10 ^bc^	−3.55 ± 0.42 ^cd^
8% (B8)	41.99 ± 0.64 ^b^	−1.84 ± 0.07 ^bc^	−3.87 ± 0.41 ^cd^
9% (B8)	44.11 ± 0.70 ^b^	−1.83 ± 0.01 ^bc^	−2.98 ± 0.52 ^cd^
10% (B10)	46.54 ± 0.96 ^a^	−2.06 ± 0.02 ^c^	−3.30 ± 0.37 ^d^

Note: Different letters in each column indicate significant differences (*p* < 0.05).

**Table 3 molecules-29-03192-t003:** Color of camellia oleogels prepared with different concentrations of GML.

Sample	L*	a*	b*
3% (G3)	23.31 ± 0.64 ^f^	−0.16 ± 0.04 ^a^	1.70 ± 0.06 ^a^
4% (G4)	25.15 ± 0.58 ^ef^	−0.18 ± 0.03 ^a^	1.64 ± 0.06 ^a^
5% (G5)	26.43 ± 1.26 ^e^	−0.24 ± 0.02 ^a^	1.32 ± 0.07 ^b^
6% (G6)	29.57 ± 1.21 ^d^	−0.48 ± 0.05 ^b^	1.13 ± 0.06 ^c^
7% (G7)	31.91 ± 0.74 ^c^	−0.84 ± 0.13 ^c^	0.94 ± 0.07 ^d^
8% (G8)	33.09 ± 0.42 ^c^	−1.06 ± 0.03 ^d^	0.82 ± 0.04 ^de^
9% (G8)	35.63 ± 0.45 ^b^	−1.28 ± 0.09 ^e^	0.78 ± 0.05 ^e^
10% (G10)	37.98 ± 1.48 ^a^	−1.84 ± 0.14 ^f^	0.72 ± 0.08 ^e^

Note: Different letters in each column indicate significant differences (*p* < 0.05).

**Table 4 molecules-29-03192-t004:** Texture characteristics of camellia oleogels with different BW contents.

Sample	Hardness/N	Springiness
3% (B3)	0.4573 ± 0.0274 ^d^	0.4025 ± 0.0134 ^a^
4% (B4)	1.2343 ± 0.0978 ^d^	0.7935 ± 0.0177 ^a^
5% (B5)	2.1713 ± 0.0839 ^d^	1.1595 ± 0.0530 ^ab^
6% (B6)	3.9406 ± 0.1054 ^c^	1.7150 ± 0.2022 ^bc^
7% (B7)	5.5390 ± 0.1464 ^bc^	2.4410 ± 0.1739 ^cd^
8% (B8)	6.9213 ± 0.6605 ^b^	2.859 ± 0.0212 ^de^
9% (B9)	9.7293 ± 1.2359 ^a^	3.6165 ± 0.1746 ^e^
10% (B10)	10.5710 ± 1.0116 ^a^	4.9595 ± 0.4957 ^f^

Note: Different letters in each column indicate significant differences (*p* < 0.05).

**Table 5 molecules-29-03192-t005:** Texture characteristics of camellia oleogels with different GML contents.

Sample	Hardness/N	Springiness
3% (G3)	0.0703 ± 0.0089 ^c^	0.0902 ± 0.0015 ^ab^
4% (G4)	0.1015 ± 0.0052 ^c^	0.1203 ± 0.0120 ^ab^
5% (G5)	0.1814 ± 0.0087 ^bc^	0.2124 ± 0.0085 ^bc^
6% (G6)	0.1505 ± 0.0450 ^bc^	0.1912 ± 0.0620 ^bc^
7% (G7)	0.3027 ± 0.0380 ^ab^	0.3135 ± 0.0180 ^cd^
8% (G8)	0.4114 ± 0.0790 ^a^	0.3621 ± 0.0341 ^d^
9% (G9)	0.4058 ± 0.0150 ^a^	0.3703 ± 0.0311 ^d^
10% (G10)	0.4521 ± 0.0121 ^a^	0.4023 ± 0.0482 ^d^

Note: Different letters in each column indicate significant differences (*p* < 0.05).

**Table 6 molecules-29-03192-t006:** Texture characteristics and color of biscuit dough with different fats.

Dough	Hardness	Springiness	Resilience	Stickiness	L*	a*	b*
Butter	21.99 ± 1.05 ^b^	0.67 ± 0.07 ^a^	0.10 ± 0.01 ^b^	11.10 ± 1.24 ^a^	76.61 ± 1.22 ^b^	0.34 ± 0.21 ^a^	19.55 ± 1.42 ^a^
6.7% GML	11.10 ± 0.55 ^e^	0.73 ± 0.14 ^a^	0.09 ± 0.01 ^b^	6.18 ± 0.78 ^b^	77.61 ± 0.49 ^b^	−0.55 ± 0.11 ^b^	17.10 ± 1.03 ^ab^
10% GML	13.79 ± 0.80 ^d^	0.69 ± 0.01 ^a^	0.47 ± 0.026 ^a^	6.49 ± 0.09 ^b^	80.82 ± 0.97 ^a^	−0.41 ± 0.26 ^b^	17.31 ± 1.71 ^ab^
6.7% BW	25.24 ± 0.38 ^a^	0.61 ± 0.03 ^a^	0.45 ± 0.04 ^a^	11.44 ± 0.92 ^a^	74.24 ± 1.13 ^c^	−0.48 ± 0.27 ^b^	15.46 ± 1.39 ^ab^
10% BW	18.36 ± 0.43 ^c^	0.58 ± 0.01 ^a^	0.42 ± 0.03 ^a^	7.80 ± 0.82 ^b^	76.22 ± 0.89 ^b^	−0.29 ± 0.38 ^b^	17.19 ± 0.53 ^b^

Note: Butter: the fat utilized in dough was butter; GML: the fat utilized in dough was camellia oleogel formed by GML at two concentrations (6.7% and 10%); BW: the fat utilized in dough was camellia oleogel formed by BW at two concentrations (6.7% and 10%). Different letters in each column indicate significant differences (*p* < 0.05).

**Table 7 molecules-29-03192-t007:** Texture characteristics, expansion ratio, and color of biscuit with different fats.

Biscuit	Hardness	Resilience	Expansion Ratio	L*	a*	b*
Butter	21.03 ± 1.24 ^c^	0.37 ± 0.06 ^a^	206.69 ± 9.72 ^bc^	75.55 ± 0.46 ^a^	2.54 ± 0.94 ^c^	25.57 ± 1.14 ^d^
6.7% GML	13.87 ± 0.38 ^d^	0.53 ± 0.16 ^ab^	326.77 ± 9.345 ^a^	76.42 ± 0.22 ^a^	0.78 ± 0.12 ^c^	22.82 ± 1.38 ^c^
10% GML	14.32 ± 1.02 ^d^	0.40 ± 0.034 ^a^	338.37 ± 16.69 ^a^	70.89 ± 0.624 ^b^	8.68 ± 1.53 ^b^	33.14 ± 1.32 ^b^
6.7% BW	24.54 ± 1.85 ^b^	0.30 ± 0.084 ^b^	215.94 ± 6.5 ^b^	60.76 ± 1.178 ^c^	14.00± 1.60 ^a^	35.65 ± 1.55 ^a^
10% BW	30.58 ± 3.17 ^a^	0.25 ± 0.068 ^b^	192.80 ± 3.76 ^c^	72.00 ± 0.207 ^b^	9.68 ± 0.92 ^b^	35.44 ± 0.97 ^ab^

Note: Butter: the fat utilized in biscuits was butter; GML: the fat utilized in biscuits was camellia oleogel formed by GML at two concentrations (6.7% and 10%); BW: the fat utilized in biscuits was camellia oleogel formed by BW at two concentrations (6.7% and 10%). Different letters in each column indicate significant differences (*p* < 0.05).

**Table 8 molecules-29-03192-t008:** Textural analysis of sausages with different fat replacements.

Fat Component	Hardness/N	Chewiness	Resilience	Springiness	Cohesiveness
Control (Lard)	11.92 ± 0.20 ^a^	6.55 ± 0.12 ^a^	0.34 ± 0.02 ^a^	0.78 ± 0.02 ^a^	0.71 ± 0.01 ^a^
Butter:Lard (1:1)	11.25 ± 0.13 ^b^	5.67 ± 0.21 ^b^	0.33 ± 0.01 ^ab^	0.73 ± 0.02 ^a^	0.69 ± 0.01 ^a^
10% BW:Lard (1:1)	10.23 ± 0.24 ^c^	5.74 ± 0.44 ^b^	0.33 ± 0.01 ^ab^	0.80 ± 0.05 ^a^	0.70 ± 0.01 ^a^
10% GML:Lard (1:1)	6.70 ± 0.31 ^e^	3.20 ± 0.36 ^d^	0.31 ± 0.01 ^b^	0.72 ± 0.05 ^a^	0.66 ± 0.02 ^b^

Note: Control group: each formulation contains 36 g of lard. Butter: each formulation contains 18 g lard + 18 g butter. BW: each formulation contains 18 g lard + 18 g 10% BW camellia oleogel. GML: each formulation contains 18 g lard + 18 g 10% GML camellia oleogel. (a–e) Different letters in the same row indicate significant differences (*p* < 0.05).

**Table 9 molecules-29-03192-t009:** Color and pH of sausages with different fats.

Samples	L*	a*	b*	pH
Lard	65.59 ± 1.49 ^a^	−1.26 ± 0.31 ^a^	13.57 ± 0.65 ^a^	5.70 ± 0.11 ^a^
Butter	62.67 ± 1.49 ^a^	−0.74 ± 0.26 ^a^	14.68 ± 1.91 ^a^	5.63 ± 0.05 ^a^
BW	62.91 ± 1.97 ^a^	−1.01 ± 0.09 ^a^	15.10 ± 2.61 ^a^	5.66 ± 0.08 ^a^
GML	65.74 ± 1.40 ^a^	−0.73 ± 0.53 ^a^	13.41 ± 2.62 ^a^	5.64 ± 0.02 ^a^

Note: Lard: each formulation contains 36 g of lard. Butter: each formulation contains 18 g lard + 18 g butter. BW: each formulation contains 18 g lard + 18 g 10% BW camellia oleogel. GML: each formulation contains 18 g lard + 18 g 10% GML camellia oleogel. Letters in the same row indicate significant differences (*p* < 0.05).

## Data Availability

All data are available in the manuscript.

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
