# Peer review of "The Impact of Beeswax and Glycerol Monolaurate on Camellia Oil Oleogel’s Formulation and Application in Food Products"

_molecules, 2024, doi:10.3390/molecules29133192_

Round 1
Reviewer 1 Report
Comments and Suggestions for Authors
Dear authors, through the following lines I will make some comments to your work.
Line 2, preparation of...it is too much in the title, maybe eliminate that part and reinforce the impact of the title in relation to all the analysis done.
Line 48, space before the parenthesis.
Line 63, the text does not have solid elements to determine that something was optimal. Optimal for what? I recommend changing the term since there is not a response surface modeling, or something related to ensure that it is "optimal". Review the conclusion and abstract about optimal oleogel.
Line 74, compare the benefits of the oil with some other oil considered to be one of the best in terms of nutrition.
If possible, justify the text of the whole article.
Line 76, table 1 please change composition to fatty acid.
Line 96, the analysis of results should be completed contrasting with the composition of beeswax and its effect on the results.
Line 113, based on?
Line 144, please first present a text section and then the table or figure to improve the readability of your article.
Tables 5 and 6, the 6.7% in beeswax is missing.
Table 5, it is necessary to discuss the results, this section is vague.
Line 202, please in each section first place a paragraph of information and then the tables and figures.
Line 226, please indicate significant statistical difference and the superscript to reference the results.
Line 207, the graph does not indicate the variable aroma.
Line 214, this is not contradictory to the term optimum. I suggest rewriting the idea.
Line 247, the results should indicate that it is percentage
Line 256, example of a table where the statistical part is not indicated, the writing of the article must be homogeneous.
Line 266, change chroma to color, chroma is more related to the use of a color meter and not to the human eye through a sensory panel.
Line 298, should indicate the details of the agitation and temperature of all stages.
Line 313, detection? means, operating variables of the experiment?
Line 331, it is not clear why they used 6.7% in the trials, it does not refer to it as random, please clarify.
Line 335, the same as line 313.
At what temperature was the dough analyzed?
For the texture tests, the biscuits were left to rest. If so, how long? Under what conditions? this should be explained
Line 349, I don't understand what you are referring to.
Line 355, please be consistent throughout the article with the terms you use in the sensory variables.
Line 392, please explain the evaluation technique used on the sausages because the results in the corresponding graph are expressed differently from the biscuits.
Author Response
1.Line 2, preparation of...it is too much in the title, maybe eliminate that part and reinforce the impact of the title in relation to all the analysis done.
Response: Thanks for your comments. We have revised the title as your suggestion.
2.Line 48, space before the parenthesis.
Response: We have corrected it.
3.Line 63, the text does not have solid elements to determine that something was optimal. Optimal for what? I recommend changing the term since there is not a response surface modeling, or something related to ensure that it is "optimal". Review the conclusion and abstract about optimal oleogel.
Response: Thanks for the comments. We have changed the description in Line 63、abstract and the conclusion part.
4.Line 74, compare the benefits of the oil with some other oil considered to be one of the best in terms of nutrition.
Response: We have reviewed relevant literature and added the fatty acid composition of olive oil for comparison.
5.If possible, justify the text of the whole article.
Response: We have added adjustment in the manuscript.
6.Line 76, table 1 please change composition to fatty acid.
Response: We have changed accordingly.
7.Line 96, the analysis of results should be completed contrasting with the composition of beeswax and its effect on the results.
Response: We have added analysis. In BW-formed oleogels, negative a* and b* values indicated a green–yellow hue, because of the inherent color of BW. This hue may be influenced by the presence of wax esters in BW, with amounts of plant pollens containing yellow-colored wax-soluble compounds
8.Line 113, based on?
Response: We have added the discussion about it. Hwang's findings suggested that a denser crystal network correlated with the in-crease of oleogel hardness, thus it was plausible that BW formed a more robust crystalline microstructure than GML.
9.Line 144, please first present a text section and then the table or figure to improve the readability of your article.
Response: Thanks for the comments. We have changed accordingly.
10.Tables 5 and 6, the 6.7% in beeswax is missing.
Response: The table has the relevant data about 6.7% in beeswax.
11.Table 5, it is necessary to discuss the results, this section is vague.
Response: We have added the discussion in the section.
12.Line 202, please in each section first place a paragraph of information and then the tables and figures.
Response: Thanks for the comments. We have changed accordingly.
13.Line 226, please indicate significant statistical difference and the superscript to reference the results.
Response: Thanks for the comments. We have changed accordingly.
14.Line 207, the graph does not indicate the variable aroma.
Response: Sorry for the mistake, we have changed aroma to color.
15.Line 214, this is not contradictory to the term optimum. I suggest rewriting the idea.
Response: Thanks for the comments. We have revised the part.
16.Line 247, the results should indicate that it is percentage
Response: Sorry for the mistake, we have revised the Y-axis in the graph.
17.Line 256, example of a table where the statistical part is not indicated, the writing of the article must be homogeneous.
Response: Thanks for the comments. We have revised all the table and graph in the manuscript.
18.Line 266, change chroma to color, chroma is more related to the use of a color meter and not to the human eye through a sensory panel.
Response: Thanks for the comments. We have changed accordingly.
19.Line 298, should indicate the details of the agitation and temperature of all stages.
Response: We have added the details of the agitation and temperature in the section. To prepare oleogels, the mixed camellia oil sample was heated at 90 °C for 120 min using a thermostatic magnetic stirrer at a rate of 500 rpm/min (S10-3, Shanghai Sile Instruments Co., Ltd., Shanghai, China) until fully dissolved. Then, each hot mixed solution was trans-ferred to a 4 °C refrigerator (BCD-601WDPR, Haier Electric Appliance, Qingdao, China) for 24 h. After that, each sample was stored in a biochemical incubator (SPX-150BSH-II, Shanghai CIMO Medical Instrument Co., Ltd, shanghai, China) at 25 °C for further re-search.
20.Line 313, detection? means, operating variables of the experiment?
Response: We have changed detection conditions to parameters.
21.Line 331, it is not clear why they used 6.7% in the trials, it does not refer to it as random, please clarify.
Response: We have made clear of the use of 6.7% in the section. Based on our prior experiments, it was observed that the oleogels formulation containing 10% (1.5g) of the gel agent exhibited significant differences in properties. So, the 10% concentration was selected for further investigation. Furthermore, to explore the effect of oleogels formed by lower gel agent concentrations on food products, a formulation containing 6.7% (1.0g) of the gel agent was also chosen for study.
22.Line 335, the same as line 313.
Response: We have changed detection conditions to parameters.
23.At what temperature was the dough analyzed?
Response: We have added the description in subsection 3.4.1, the dough was analyzed at 25°C.
24.For the texture tests, the biscuits were left to rest. If so, how long? Under what conditions? this should be explained
Response: Thanks for the comments. We have explained the storage conditions in this part. The cooled biscuit samples were stored in hermetically sealed bags for evaluation at room temperature for 0–3 days.
25.Line 349, I don't understand what you are referring to.
Response: We have revised the section. In color measurement of dough and biscuits, we have added the details. Prior to measurement, the device was calibrated, and the prepared dough samples were shaped into uniform sheets using a mold, with dimensions of 5 cm in length, 4 cm in width, and 1 cm in thickness. Similarly, the biscuits, once cooled to room temperature, were also measured for their color attributes. Each sample was analyzed in triplicate to ensure the reliability of the results.
26.Line 355, please be consistent throughout the article with the terms you use in the sensory variables.
Response: Thanks for the comments. We have revised the terms.
27.Line 392, please explain the evaluation technique used on the sausages because the results in the corresponding graph are expressed differently from the biscuits.
Response: Thanks for the comments. We have added the evaluation technique in the section. Post-cooking, the samples were presented to the participants immediately, each labeled with a unique random number. Prior to the sensory analysis, the objectives and method of the evaluation were introduced, including the use of standardized scales. The sensory at-tributes assessed included appearance, color, flavor, taste, and organizational status of the sausage samples. Each sensory attribute was evaluated using a 20-point scale, contrib-uting to a cumulative total of 100 points for the overall sensory evaluation.
Reviewer 2 Report
Comments and Suggestions for Authors
The manuscript on "Preparation of Camellia Oil Oleogel and its Application in Biscuits and Sausages" is an experimental article, clearly and well structured. The authors analysed the fatty acid composition of camellia oil and used it as a base with gelling agents (beeswax/glycerol monolaurate) to create oleogel systems of different concentrations. The colour, texture and structure of camellia oil oleogels were analysed to determine the optimal formulation parameters for fat replacement in biscuits and sausages. The results show that beeswax is a more acceptable fat replacement in the form of an oleogel in biscuits. In general, it is an alternative to lard to maintain the desired textural properties of the sausage, while glycerol monolaurate needs to be optimised.
There are several comments to the article:
1. In subsection 3.1, all chemical reagents used in this study should be listed.
2. In subsection 3.2, describe in detail the preparation of camellia oil samples for further chromatographic analysis. What column and carrier gas were used for chromatography? State the volume of the injection and the number of replicates of the experiment.
3. Describe in detail in 3.3.1 the method of colourimetric analysis of camellia oil oleogels. Was the colourimetric measurement of colour performed after several days of storage of the samples? Was there a change in colour?
4. Literary references should be cited in accordance with the requirements of the journal and errors in this list of references should be corrected.
This manuscript is scientifically sound, contains 44 references, and is marked by relevance and novelty. The conclusions are consistent with the evidence and arguments presented. The ethical and data availability statements are adequate.
In my opinion, the manuscript can be accepted for publication in the journal Molecules after minor additions and corrections.
Author Response
1.In subsection 3.1, all chemical reagents used in this study should be listed.
Response: Thanks for the comments. We have added the other chemical reagents used in materials section.
2.In subsection 3.2, describe in detail the preparation of camellia oil samples for further chromatographic analysis. What column and carrier gas were used for chromatography? State the volume of the injection and the number of replicates of the experiment.
Response: Thanks for the comments. We have added the details accordingly, the column was Agilent DB-23 column(30m×0.25mm×0.2μm), carrier gas was helium at the flow of 1mL/min, the injection volume was 1μL.
3.Describe in detail in 3.3.1 the method of colourimetric analysis of camellia oil oleogels. Was the colourimetric measurement of colour performed after several days of storage of the samples? Was there a change in colour?
Response: We have added the details of the method of colourimetric analysis. We haven’t assessed the color of oleogels after several days of storage. In our future study, the storage time will be extended to explore the effect of gels on storage characteristic in oleogels.
4.Literary references should be cited in accordance with the requirements of the journal and errors in this list of references should be corrected.
Response: Thanks for your comments. We have carefully checked the whole references section and revised it accordingly.
Round 2
Reviewer 1 Report
Comments and Suggestions for Authors
Dear authors, I appreciate that my review has been taken into account for the improvement of your research article. At the moment I consider that it is in conditions to be publishable material. I congratulate your effort and dedication to contribute to food science and technology in the search for food alternatives of lower caloric impact and better for cardiovascular health.